# The Role of Vitamin D Supplementation in Children with Autism Spectrum Disorder: A Narrative Review

**DOI:** 10.3390/nu14010026

**Published:** 2021-12-22

**Authors:** Monia Kittana, Asma Ahmadani, Lily Stojanovska, Amita Attlee

**Affiliations:** 1Department of Nutrition and Health, College of Medicine and Health Sciences, United Arab Emirates University, Al Ain P.O. Box 15551, United Arab Emirates; 202090016@uaeu.ac.ae (M.K.); 201270079@uaeu.ac.ae (A.A.); lily.stojanovska@uaeu.ac.ae (L.S.); 2Institute for Health and Sport, Victoria University, Melbourne, VIC 8011, Australia

**Keywords:** autism spectrum disorder, vitamin D deficiency, vitamin D supplement, ASD severity, children

## Abstract

Children with autism spectrum disorder (ASD) present with persistent deficits in both social communication and interactions, along with the presence of restricted and repetitive behaviors, resulting in significant impairment in significant areas of functioning. Children with ASD consistently reported significantly lower vitamin D levels than typically developing children. Moreover, vitamin D deficiency was found to be strongly correlated with ASD severity. Theoretically, vitamin D can affect neurodevelopment in children with ASD through its anti-inflammatory properties, stimulating the production of neurotrophins, decreasing the risk of seizures, and regulating glutathione and serotonin levels. A Title/Abstract specific search for publications on Vitamin D supplementation trials up to June 2021 was performed using two databases: PubMed and Cochrane Library. Twelve experimental studies were included in the synthesis of this review. Children with ASD reported a high prevalence of vitamin D deficiency or insufficiency. In general, it was observed that improved vitamin D status significantly reduced the ASD severity, however, this effect was not consistently different between the treatment and control groups. The variations in vitamin D dose protocols and the presence of concurrent interventions might provide an explanation for the variability of results. The age of the child for introducing vitamin D intervention was identified as a possible factor determining the effectiveness of the treatment. Common limitations included a small number of participants and a short duration of follow-ups in the selected studies. Long-term, well-designed randomized controlled trials are warranted to confirm the effect of vitamin D on severity in children with ASD.

## 1. Introduction

Autism spectrum disorder (ASD), a common and rapidly growing neurodevelopmental disorder [1], is characterized by persistent deficits in social communication and social interaction along with restricted and repetitive behaviors in the early developmental period, causing significant impairment in important areas of functioning [2]. A WHO report (2019) estimated the worldwide prevalence of ASD to be 1 in 160 children [3]. In the United States, the prevalence has increased from 1 in 59 children in 2014 to 1 in 54 children in 2020 [4]. This increase may be due to either improved surveillance and diagnosis, or a true increase in prevalence [1,5]. It is estimated that 10–20% of ASD cases are attributed to genetic causes [6]. On the other hand, possible environmental etiologies include advanced parental age, lead exposure, caesarean section, air pollution, and maternal obesity, hypertension, and/or diabetes [5,7,8,9,10].

Compared to an adult, the young brain is profoundly metabolically active and is responsible for approximately 60% of the total body energy consumption [11]. It is well-established that early nutrition is critical for neurodevelopment [12], rendering it a cost-effective approach for the prevention of mental health problems [13]. Deficiencies of key nutrients at critical periods including protein, long-chain polyunsaturated fatty acids, minerals such as zinc and iron, and vitamins (vitamins A, D, B_6_, B_9_, and B_12_) [11,13,14], can compromise brain function due to their role in signaling cascades that affect the neuronal functional capacity [11]. Based on the data from epidemiological studies, vitamin D deficiency specifically has been hypothesized to increase the risk of ASD [1].

The association between vitamin D and ASD emerged with increased evidence of higher ASD prevalence among children who lived in areas with low ultraviolet-B rays in comparison to those residing in sunny areas [1]. Vitamin D is physiologically converted to its active form, 1,25(OH)D (Calcitriol), through two consecutive hydroxylation processes in the liver and kidneys [15]. Calcitriol is a neuroactive hormone [12], that is responsible for different aspects of brain development and early cognitive development [1,16]. Vitamin D assists in neural cell proliferation and neurotransmission functions [16,17], thus theoretically affecting the neurodevelopmental processes.

Further, vitamin D plays an important role in the modulation of the inflammation system by regulating the production of inflammatory cytokines and immune cells, which are crucial for the pathogenesis of many immune-related diseases [18]. Strong inflammation states are associated with ASD, attributed to the link of elevated inflammatory cytokines with cognitive impairment [1]. Significantly higher concentrations of interleukin (IL)-1β, IL-6, IL-8, interferon-gamma (IFN-γ), eotaxin, and monocyte chemotactic-1 proteins (MCP-1), along with significantly lower transforming growth factors-β1 were reported in individuals with ASD compared to their healthy controls [19]. Elevated IL-6 and TNF-α in children with ASD were positively correlated with ASD severity as measured by the Childhood Autism Rating Scale (CARS) (R = 0.638 and R = 0.699, respectively (*p* < 0.0001)), and are also used as biomarkers of ASD diagnosis [20]. Animal-model studies revealed that elevated IL-6 levels in mice disrupt the balance of synaptic transmissions, and mediate autistic-like behaviors including decreased social interaction, impaired cognitive abilities, and learning deficits [21]. Vitamin D status was reported to exert a negative association with inflammation, as summarized in a systematic review on typically developing (TD) children and adolescents [22]; while another systematic review of immune cell studies reported consistent observations of suppression in MCP-1, IL-6, and IL-8 with vitamin D supplementation [23].

Children with ASD are commonly reported with deficient levels of vitamin D and disrupted low serotonin levels in the brain [24]. Serotonin levels are implicated in regulating brain functions including neurogenesis [25]. Both, experimental tryptophan and serotonin depletion, manifest as behavioral characteristics similar to ASD [24]. Vitamin D inadequacy reduces serotonin concentrations in the brain by decreasing the synthesis of tryptophan hydroxylase 2 (TPH2), thus impacting the structural and neuronal wiring of the brain [24]. Individuals with ASD may present with a phenomenon known as serotonin anomaly, characterized by high serotonin in peripheral blood cells, possibly due to increased gut serotonin production, but lower concentrations in the brain [24]. Brain serotonin dysregulation is critical for neuropsychiatric illnesses, including ASD, considering its impact on executive function, sensory gating (preventing an overload of irrelevant information), impulsive aggression towards self and others, and antisocial behavior [26]. Polymorphisms in the TPH2 gene may contribute to reduced brain serotonin synthesis [24] and have been associated with stereotyped behavior patterns in ASD [27]. As indicated earlier, vitamin D is hypothesized to increase TPH2 gene expression, and thus, increase the levels of serotonin synthesis in the brain [28] that can positively promote social behavior and influence emotional social cues. Serotonin depletion in neonatal mice showed similar behavioral characteristics to autism [24,29,30].

Vitamin D is also involved in enhancing glutathione peroxidase 1 (GP1) levels [31], thus reducing oxidative stress [31,32]. Glutathione redox imbalance contributes to ASD as it increases the expression of pro-inflammatory cytokines and can significantly impact neuro-inflammation [32]. Additionally, vitamin D may stimulate the production of neurotrophins, such as Nerve Growth Factor (NGF) and Glial cell-derived Neurotrophic factor (GDNF) [33,34], which are proteins involved in neuronal development and function [35]. Vitamin D is shown to increase NGF levels in animals [33], in-vitro [36] and in human studies [37], and specifically in children with ASD [38]. A very potent increase of GDNF in-vitro studies suggests the in-vivo regulation of GDNF [34]. However, the modulatory effect of these neurotrophins in ASD is not clear, as evident in one meta-analysis [39], wherein NGF levels were found to be significantly higher in children with ASD compared to controls, and another recent study reported high levels in all children with ASD [40], propounding a critical role of NGF in ASD development [39]. A major limitation in these studies is the small sample size, warranting the need for robust studies on the role of neurotrophins in ASD, and in relation to vitamin D status. Although the above are some of the biological systems that are thought to be altered in ASD, the differences between children with ASD and TD children are inconsistent [41]. Nonetheless, vitamin D has shown to upregulate neurotrophins expression [42,43], thereby exerting beneficial effects on the developing brain; however, experimental evidence in relation to children with ASD is scarce.

Vitamin D is also reported to increase the seizure threshold, thereby assisting in controlling the occurrence and severity of seizures in children [42,44]. This is relevant in context to ASD which is often accompanied by many co-morbidities [45], epilepsy being one of the most common conditions [46].

Both vitamin D deficiency and insufficiency, are commonly reported among healthy children of different populations [47,48]. Children with ASD consistently report significantly lower vitamin D levels than TD children [17,42,49,50,51,52,53,54,55]. Moreover, vitamin D deficiency is shown to be strongly correlated with ASD severity [56]. Sun exposure in children with ASD was significantly lower than in TD children, which may provide an explanation to this difference [51], although heritable vitamin D deficiency may be another factor for the compromised vitamin D status, as vitamin D-related enzymes, receptors, and binding proteins are all under genetic control [42].

Early monitoring and comprehensive treatment are critical to the symptoms-severity control in children with ASD [57]. Therefore, exploring whether the experimental evidence supports this theoretical role of vitamin D supplementation in children with ASD is important in order to design early interventions that may provide promising solutions to reducing the severity in children with ASD. Accordingly, this article aims to review the evidence from experimental studies and evaluate the effectiveness of vitamin D supplementation in attenuating the severity in children with ASD.

## 2. Materials and Methods

A Title/Abstract specific search was performed using two databases: PubMed and Cochrane Library. The search included publications from any date up to June 2021. Additionally, the reference lists of the identified trials were hand-searched. The search terms used included autism-related terms: (‘Autism’ OR ‘Autism, early infantile’ OR ‘Autism, infantile’ OR ‘Kanner’s syndrome OR Autistic’ OR ‘Autistics disorder’) and vitamin D, and a prospective experimental study design using the search terms ‘trial’ OR ‘experiment’. The study designs included randomized control trials (RCT), non-randomized clinical trials, quasi-experimental designs, case studies, and case series. Additional inclusion criteria were (1) a pediatric population, with ages less than 18 years, (2) inclusion of a supplementation group for vitamin D only, and (3) availability of a full-text article. Through the search, 59 relevant studies were identified. After screening the abstracts against the inclusion criteria and removing duplicates, a total of ten studies were accepted. Two additional articles were identified through the reference lists of other articles. Therefore, a total of 12 studies were included in this review.

## 3. Effectiveness of Vitamin D Supplementation in Children with ASD

### 3.1. Vitamin D Status and Association with ASD Severity

Table 1 and Table 2 provide an overall summary of the 12 studies and the main outcomes in regards to ASD severity measures. The reviewed studies reported a high prevalence of vitamin D deficiency in children with ASD ranging from 13% to 100% in the selected samples of children with ASD [55,58,59,60]. Vitamin D deficiency is defined as levels below 20 ng/mL, while those with less than 10 ng/mL are diagnosed with severe deficiency, and levels between 20–30 ng/mL are referred to as vitamin D insufficiency [61]. The studies reported baseline levels of 23.6 ± 13.3 ng/mL [62], 21.68 ng/mL [63], and 10.84 ± 16.80 ng/mL and 8.19 ± 6.78 ng/mL, respectively, in the treatment and placebo groups of children with ASD [58]. The other studies only recruited vitamin D deficient children [64,65,66,67,68,69].

Saad et al. (2015) reported significantly lower 25(OH)D levels in children with severe ASD compared to children with moderate/mild ASD [55]. Similarly, Feng et al. (2017) found a significant negative correlation between vitamin D status and the total score for psychiatric symptoms and behavioral disturbances as measured by the Aberrant Behavior Checklist (ABC) scores in children with ASD [60]. These studies highlighted the relationship between vitamin D status and ASD severity. Table 3 summarizes the effectiveness of vitamin D supplementation on ASD severity measured using different questionnaire-based assessment tools.

### 3.2. Effectiveness of Vitamin D Supplementation on ASD Severity

A trend of significantly improved outcomes based on several ASD severity scales was observed with vitamin D_3_ supplementation, indicating a plausible role of vitamin D, especially in a population with a high prevalence of vitamin D deficiency. As evident in Table 1, Table 2 and Table 3, a consistent improvement was observed across the trials in the Childhood Autism Rating Scale (CARS) [55,58,60,62,64,65] and Autism Treatment Evaluation Criteria (ATEC) [58,62], in some subscales of ABC and Autism Behavior Checklist (ABC*) [55,58,59,60,69], and others, except for two studies [63,67]. These outcomes necessitate a careful interpretation of the data, considering the limitations of the small sample size of children with ASD and the possible effect of other confounding factors, some of which are sunlight exposure, ethnicity and skin color, parathyroid disorders, dietary habits, and nutritional status [70,71].

Although some promising results were noted in the before–after effect of vitamin D supplementation on ASD severity, significant differences were not consistently observed between the groups. Therefore, these results remain inconclusive in establishing the independent effectiveness of vitamin D, attributed to certain limitations, including the presence of concurrent treatments and significant differences in baseline serum vitamin D levels between cases and controls. However, among the trials that did not present with these limitations, two showed favorable results [58,66], and three demonstrated limited or no favorable effects [63,68,69]. The main difference in the study protocols of these trials was the vitamin D supplementation approach.

It is challenging to attribute the improvement in ASD severity scales to vitamin D supplementation in the presence of other exposures. This limitation may explain the non-significant differences in some studies [59,62]. Azzam et al. (2015) showed significant improvements in CARS and ATEC scales in both supplemented and non-supplemented groups who attended behavioral and speech therapy sessions for 30 min, three times/week [48]. Moreover, the non-supplemented group initially had higher vitamin D levels, and therefore, at post-treatment, both groups reached similar levels, possibly leading to no significant differences in the main outcomes [62]. Similarly, the control group in the Ucuz et al. (2015) study also had baseline normal vitamin D levels, compared to deficient levels in the treatment group, which could ultimately mask the difference in improvement and explain the lack of significant differences between groups, despite the significant change within the treatment group. Additionally, one trial reported no differences within the treatment group [63], contradicting the reported outcomes in other studies [58,66]. Postulated reasons were the advanced age of the child or the fixed dosing.

Generally, two different approaches were adopted for vitamin D supplementation: either a fixed dose [59,60,62,63,64,65,68,69], usually of 2000 IU/day [62,63,68,69], or body weight-dependent dose, set at 300 IU/kg body weight/day [55,58,66]. In an analytical study of 22,214 records of vitamin D supplementation, the authors concluded that bodyweight-specific recommendations should be implemented to achieve serum vitamin D targets [72]. In fact, the current RDA of vitamin D is limited, as it still recommends 600 IU/day for young adults regardless of their body weight [73]. Further, serum vitamin D levels were lower by 8.0 and 19.8 nmol/L in overweight and obese participants, respectively, compared to their normal counterparts following vitamin D supplementation protocols [72]. Children with sickle cell disease who received higher vitamin D doses per weight had higher serum vitamin D levels following three months of supplementation, indicating the limitation of using fixed doses [74]. Nonetheless, weight-based doses were not consistently superior to the fixed doses [75], and therefore may not be solely attributed to the results lacking statistical significance.

Another factor that may influence the interpretation of the effect of vitamin D supplementation is the endpoint serum vitamin D levels. Although Mazahery et al. (2019) concluded that vitamin D did not have a significant impact on the Social Responsiveness Scale (SRS)-2 or Sensory Processing Measure (SPM) scales, subgroup analysis for vitamin D showed greater improvements in SRS-total and SPM-social participation subscale in children with endpoint levels of 25(OH)D > 100 nmol/L compared to those with levels ≤100 nmol/L (*p* ≤ 0.05) [68].

In trials without control groups, consistently favorable outcomes of vitamin D supplementation were reported despite differences in sample size, vitamin D dose, and duration of intervention [39,46,48,50]. Interestingly, the ASD severity reflected in poor CARS and ABC* measures worsened with the discontinuation of vitamin D supplementation, indicating that continuous supplementation is critical for maintaining the therapeutic effect of vitamin D [65]. Despite the favorable effects of vitamin D, these results remain non-generalizable due to the limitations of study design, yet drawing inferences towards the potential role of vitamin D in ASD severity.

An overall summary of vitamin D supplementation in attenuating ASD severity and postulated mechanisms is presented in Figure 1, which requires further investigation in order to establish the clear role of vitamin D in alleviating ASD severity.

### 3.3. Vitamin D Supplementation and Age for Intervention

Developmental processes are under precise temporal control, and these should be considered for the timing of the treatment, especially for neurodevelopmental disorders such as ASD [76]. It is suggested that there is a window of opportunity in which vitamin D supplementation can be more effective. This was highlighted by Feng et al. (2017) in a subgroup analysis based on age, who reported a significantly higher reduction in both CARS and ABC scores in children of three years or younger (*n* = 20) in comparison to those older than three years (*n* = 17) [60]. Kerley et al. (2017), who reported no significant effects of vitamin D supplementation, attributed it to the advanced age of participants (inclusion criteria: <18 years), at which the neuronal networks would have already been established and the maximum benefit of the supplementation could not be accrued [63]. Other trials with recruitment criteria of relatively younger ages reported better improvements compared to the control groups [62,66,69].

The introduction of Vitamin D at much earlier stages during the prenatal period may also be associated with a lower risk of ASD as highlighted in a birth cohort of 4334 individuals, wherein vitamin D deficiency at mid-gestation increased the odds of ASD (Odds ratio: 2.42; 95% CI: 1.09 to 5.07) [77]. Another recent meta-analysis reported that reduced maternal or neonatal vitamin D was significantly associated with increased odds of ASD development (OR: 1.54, 95% CI: 1.12; 2.10) [78]. In a small trial of 19 pregnant mothers who already had a child with ASD, vitamin D supplementation during gestation and vitamin D prescriptions for the new infant resulted in a lower recurrence rate of ASD [79].

### 3.4. Effectiveness of Vitamin D Supplementation on Secondary Outcomes

Two studies reported the effect of vitamin D on inflammatory biomarkers in ASD. To elaborate, Kerley et al. (2017) reported no differences in mean C-Reactive Protein (CRP) levels between the supplementation and the placebo groups [63], while Javadfar et al. (2020) reported a trend of decreased IL-6 levels following vitamin D supplementation (−1.8 ng/mL, *p* = 0.08); however, this was not statistically different compared to the placebo group (*p* = 0.527) [58]. CRP has been previously found to be higher in children with ASD compared to TD children that increased with ASD severity [80]. This effect of vitamin D on CRP is inconsistent in the literature [81,82,83,84]. The vitamin D dosage and duration of the interventions may explain the heterogeneity in results across studies [84]. IL-6 was hypothesized to decrease due to the anti-inflammatory trait of vitamin D; however, only a trend of decreased levels was observed in children with ASD (*p* = 0.082), with no difference between groups [56]. The dysregulated production of IL-6 can contribute to chronic inflammation and autoimmunity [85]. To date, studies on the effect of vitamin D on IL-6 in populations with ASD have not been published [58]. However, in-vitro studies demonstrated its effect on decreasing IL-6 concentration [86,87], in a dose-dependent manner [86]. Previously, vitamin D significantly reduced IL-6 levels in-vivo studies in adults with multiple sclerosis and ventilator-associated pneumonia [88,89]. On the other hand, some reports showed no effect of vitamin D on IL-6 levels [90,91], expected due to the variations in the study designs, dosage, and duration of vitamin D supplementation [90].

Regarding neurotrophins, Ucuz et al. (2014) reported that NGF levels were significantly increased in the vitamin D supplementation group (baseline: 24 ± 27 pg/mL^−1^, endpoint: 59 ± 53 pg/mL^−1^; *p* = 0.04), although GDNF levels did not show any improvement (*p* = 0.73) [59]. Javadfar et al. (2020) also reported a trend of decreased serotonin levels (−4.07 ng/mL, *p* = 0.085) in ASD, albeit not significant, compared to the placebo group (*p* = 0.126) [58]. As previously discussed, NGF and GDNF are recognized for their effects on nervous system development and maintenance [92,93]. NGF is hypothesized as one of the mediators of positive effects of vitamin D in ASD [59]. Further, NGF was significantly correlated with hyperserotonemia in patients with ASD, although were not significantly correlated with ASD severity [40]. Previous studies hypothesized the role of vitamin D in the regulation of TPH1 and TPH2, implicated in the pathway of ASD [24]; however, evidence from in-vivo trials of ASD is missing.

### 3.5. Safety and Tolerability of Vitamin D Supplementation

Vitamin D supplementations are considered to be safe as the reviewed trials did not report adverse effects. However, a few transient side-effects were cited, including mild cases of skin rashes, itching, and diarrhea, and the children continued with the supplementation [55]. Three patients dropped out of the study due to reported hyperactivity and reduced sleep (*n* = 2) and diarrhea (*n* = 1) [62]. In the vitamin D + omega-3 trials, one child reported an allergic reaction [68]. Five trials reported no adverse side effects [58,60,63,65,67], and three trials did not report the safety or tolerability of vitamin D [59,64,66]. These are in line with results of other vitamin D supplementation trials in children that concluded safety in terms of the low incidence of adverse effects simultaneously with elevated vitamin D levels on different doses of vitamin D, including 2000 IU/day [94], or 7000 IU/day [95]. A minimum of 40 ng/mL serum vitamin D levels are optimal to obtain benefits among individuals with ASD [55]. A serum level of up to 220 nmol/L(88 ng/mL) is generally considered safe [96].

## 4. Limitations

The findings from the 12 trials should be interpreted cautiously as several factors limit the generalizability of these results. The most common drawbacks were the small sample size and insufficient study power, leading to the lack of sample representativeness. Experimental controlled trials would be the most appropriate study design, yet out of the 12 studies analyzed, only 6 were RCTs. Sampling bias, lack of control groups, lack of blinding from the outcomes between the groups in some trials, high dropout rates, and loss to follow-up were the major limitations that could affect the internal and external validity of the studies. From the reviewed literature on vitamin D trials, it is obvious that it may be challenging to control for other confounding factors such as sunlight exposure, outdoor physical activity, and dietary intake of vitamin D, geographical location, the season of conducting the study, and any additional intake of vitamin/mineral supplementations can increase the heterogeneity of the results. There was also a variation in the tools used to assess ASD symptom-severity. Although the broad outcome of each tool is to report ASD severity, each tool is comprised of different subscales and items that address specific aspects of ASD-related behaviors; thereby possibly leading to variations in the results.

Besides this, heterogeneity in the results is also attributed to the single nucleotide polymorphisms (SNPs) in the vitamin D receptor (VDR). VDR SNPs have also been suggested to play a role in ASD; VDR rs2228570 (*FokI*) polymorphism specifically, showed a strong correlation with ASD and hyperactivity behavior, increased risk of giving birth to children with ASD in mothers, and showed a significant difference between healthy siblings and children with ASD [97]. Although not explored in the reviewed trials, the benefits of vitamin D supplementation may be under the influence of the vitamin D receptor genotype, which mediates the biological activity of vitamin D [97]. For example, Barry et al. (2017) reported that the effectiveness of vitamin D supplementation in patients with adenomas varied significantly based on 2 VDR SNPs rs7968585 and rs731236 (*TaqI*) [98]. This requires verification studies in children with ASD, to investigate the effectiveness of vitamin D supplementation in context to VDR SNPs, and if individuals with a higher functional loss of VDR would benefit from different dosages of supplements. Additionally, SNPs in the vitamin D binding protein (DBP) coding protein have been established as determinants of circulating levels of 25(OH)D [99]. In healthy infants and toddlers, two SNPs (rs7041 and rs4588) significantly affected both circulating DBP and 25(OH)D [99]. Further, variation in the SNP-rs4588 was associated with increased ASD risk in children [100].

Better results could be achieved if different metabolites of vitamin D were tested as level biomarkers and if different vitamin D supplementation titers were used to check and compare the efficacy of the trial at different vitamin D concentrations. Due to the inconclusive findings on the markers linking vitamin D and ASD severity, it is proposed that large-scale and multi-centric RCTs are conducted for establishing strong evidence on the effect of vitamin D supplementation on ASD severity.

## 5. Conclusions

Due to the active role of vitamin D in the neurodevelopment of the fetus, it has a potential role in influencing the progression of neurodevelopmental disorders, including ASD. The data from the reviewed experimental studies provide supportive evidence to the role of vitamin D in reducing the severity of ASD as evidenced by the significant changes of the ASD severity measures. However, these effects are inconclusive due to the inherent limitations of reviewed trials. In general, the studies included few participants and interventions were of short duration. Additionally, ASD severity assessment tools were not consistent among studies. Nevertheless, vitamin D supplementation may still be considered under medical supervision due to the high prevalence of baseline vitamin D deficiency in children with ASD, potential effects on ASD severity, and the evidenced safety and tolerability of the supplements across all trials. However, longer, and more robust trials are needed to confirm whether vitamin D can be introduced as an adjunct therapeutic measure to attenuate the ASD severity.

## Figures and Tables

**Figure 1 nutrients-14-00026-f001:**
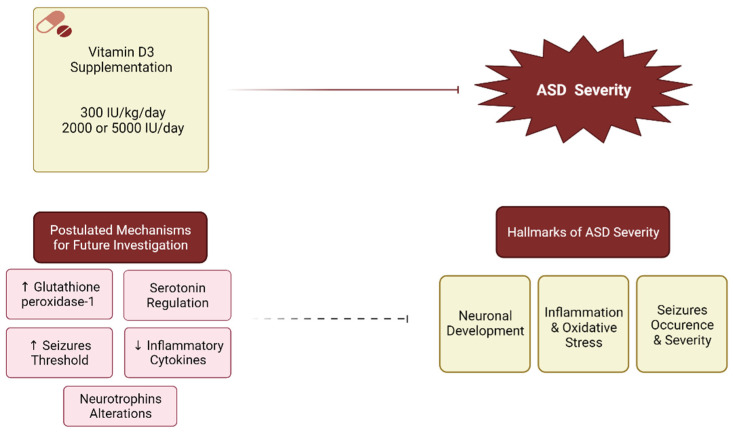
Vitamin D supplementation in attenuating ASD severity and postulated mechanisms. Inhibition: 
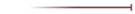
, Postulated inhibition: 
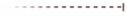
.

**Table 1 nutrients-14-00026-t001:** Human clinical trials with control groups of vitamin D and ASD severity.

#	First Author (Year)	Study Design	Total Participants Analyzed, Age, and Gender	Treatment Details and Total Participants	Control Group/s and Total Participants	Duration	25(OH)D Change in Cases (ng/mL)	25(OH)D Change in Controls (ng/mL)	ASD Severity Measure	Before-after Change (Mean Trend)	Between Groups Comparison (*p*-Value)
1	Azzam (2015) [62]	Prospective, randomized, controlled study	*n* = 21 Age: 2–12 y. M:F: 16:5	2000 IU/day of vitamin D_3_ (*n* = 10)	No supplement (*n* = 11) TD children (*n* = 23)	6 months	47 ± 20 to 71 ± 35 (nmol/L)	69 ± 41 to 70 ± 36 (nmol/L)	CARS	Decrease †	NS *
									VABS	Increase †	NS *
									ATEC	Decrease †	NS *
2	Javadfar (2020) [58]	Randomized, double-blind, placebo-controlled, parallel-group trial	*n* = 43 Age: 3–13 y. M:F: 36:7	300 IU/kg/day (Max. 6000 IU/day) of vitamin D_3_ (*n* = 22)	Placebo (*n* = 21)	15 weeks	8.19 ± 6.78 to 39.10 ± 33.71 †	10.84 ± 16.80 to 8.94 ± 8.03	CARS	Decrease †	S
									ATEC	Decrease †	S
									ABC-C subscales	Irritability: Decrease † Lethargy: Decrease † Others: Decrease	NS
3	Kerley (2017) [63]	double-blind, randomized, placebo-controlled trial.	*n* = 38. Age: <18 y. M:F: 33:5	2000 IU vitamin D_3_ (*n* = 18)	Placebo (*n* = 20)	20 weeks.	58.4 ± 17.9 to 86.1 † (nmol/L)	51.7 ± 19.8 to 50.6 (nmol/L)	ABC	Decrease	NS
									SRS	Decrease	NS
									DD-CGAS	Increase in all subscales	Self-care subscale: S, other: NS
4	Mazahery (2019) [69]	Randomized placebo controlled double-blind study	*n* = 73 Age: 2.5–8 y. M:F: 60:13	2000 IU/day vitamin D_3_ (*n* = 19)	omega-3 (*n* = 23) Vitamin D3 and omega 3 (*n* = 15) Placebo (*n* = 16)	12 months	68 ± 21 to an increase by 27 ± 14 (nmol/L)	55 ± 27 to an increase by 38% (nmol/L)	ABC domains	Irritability and hyperactivity: Decrease † Other: Decrease	Irritability and hyper-activity: S * Others: NS *
5	Mazahery (2019) [68]	Randomized placebo controlled double-blind study	*n* = 73 Age: 2.5–8 y. M:F: 60:13	2000 IU/day vitamin D_3_ (*n* = 19)	omega-3 (*n* = 23) Vitamin D3 and omega 3 (*n* = 15) Placebo (*n* = 16)	12 months	68 ± 21 to an increase by 27 ± 14 (nmol/L)	55 ± 27 to an increase by 38% (nmol/L)	SRS	Decrease	NS *
									SPM	Decrease	NS *
6	Moradi (2020) [66]	Randomized Controlled Trial	*n* = 100 Age: 6–9 y. M:F: 100:0	300 IU/kg/day (max. 5000 IU/day) of vitamin D_3_ (*n* = 25)	perceptual-motor exercises (*n* = 25) Exercises and Vitamin D_3_ (*n* = 25) Placebo (*n* = 25)	3 months.	4.61 ± 12.60 to 6.48 ± 24.36 †	4.87 ± 11.52 to 3.95 ± 11.08	SS-GARS-2	Decrease †	S *
7	Ucuz (2014) [59]	Before-and-after study	*n* = 21 Age: 2–5 y.	Vitamin D3 dose (*n* = 11) - 25(OH)D < 15: 5000 IU/day - 25(OH)D 15–20: 400 IU/day - 25(OH)D is 15–20: Continue	No supplementation (*n* = 10)	4 months. Follow-up at 6 months	<20 (exact before-after values NR)	≥20 (exact before-after values NR)	ABC *	Decrease †	NS
									Denver II:	Increase †	NS

NR: Not Reported. †: *p* < 0.05. S: Significant. NS: Not Significant. N/A: Not Applicable. m.: month. y.: year. M: Male. F: Female. TD: Typically Developing. 25(OH)D: 25-hydroxyvitamin D₃. IU: International Unit. CARS: Childhood Autism Rating Scale. ABC: Aberrant Behavior Checklist. ABC-C: ABC-Community. ABC *: Autism Behavior Checklist. SRS-2: Social Responsiveness Scale-2, SPM: Sensory Processing Measures, ATEC: Autism Treatment Evaluation Criteria. VABS: Vineland Adaptive Behavior Scale. GARS-2: Gilliam Autism Rating Scale-Second Edition. Denver-II: Denver Developmental Screening Test II. DD-CGAS: Developmental Disabilities—Children’s Global Assessment Scale. * Between-group comparisons is between vitamin D and placebo in studies with multiple control groups.

**Table 2 nutrients-14-00026-t002:** Human clinical trials without control groups of vitamin D and ASD severity.

#	First Author (Year)	Study Design	Total Participants Analyzed, Age, and Gender	Treatment Details and Total Participants	Duration	25(OH)D Levels (ng/mL) Cases	ASD Severity Measure	Before-after Change (Mean Trend)
1	Bent (2017) [67]	Open-Label Trial	*n* = 3 Age: 3–8 y. M:F: 0:3	6000 IU/day of vitamin D_3_ oil drops for the first 10 days followed by 300 IU/kg/day	3 months	28 ± 2 to 82 ± 50	SRS	Decrease †
							ABC	Decrease
2	Feng (2017) [60]	Before-and-after study design	*n* = 37 Mean age: NR ** M:F: NR	150,000 IU/month IM, and 400 IU/day orally of vitamin D_3_	3 months	Significant increase (values could not be interpreted from figure)	ABC *	Decrease †
							CARS	Decrease †
							ATEC	Decrease †
							ABC-C subscales	Irritability: Decrease † Lethargy: Decrease † Others: Decrease
3	Jia (2014) [64]	Case Report	*n* = 1 Age: 32 m. M:F: 1:0	150,000 IU/month IM and 400 IU/day orally of vitamin D_3_	2 months	12.5 to 81.2	ABC *	Decrease
							CARS	Decrease
							CGI	Decrease
4	Jia (2019) [65]	Case Series	*n* = 3 Age: 19–48 m. M:F: 3:0	Patients 1: 150,000 IU/month IM and 800 IU/day orally of vitamin D_3_. Patients 2 and 3: NR	1 month (repeated twice)	1st: 14.50 ± 5.39 to 46.70 ± 6.78 2nd: 28.10 ± 0.78 to 45.17 ± 3.04 •	ABC *	Decrease
							CARS	Decrease
5	Saad (2016) [55]	Open label trial	*n* = 83 Age: 3–9 y. M:F: 83:0	300 IU/kg/day (max.: 5000 IU/day) of vitamin D_3_	3 months	NR	CARS	Decrease †
							ABC subscales	Irritiability, lethargy, hyperactivity, stereotypic behavior: Decrease † Inappropriate speech: Decrease

NR: Not Reported. †: *p* < 0.05. m.: month. y.: year. M: Male. F: Female. 25(OH)D: 25-hydroxyvitamin D₃. IU: International Unit. CARS: Childhood Autism Rating Scale. ABC: Aberrant Behavior Checklist. ABC-C: ABC-Community. ABC *: Autism Behavior Checklist. ATEC: Autism Treatment Evaluation Criteria. • Means values were calculated based on the values presented in the study. ** The sample was taken from a larger group with a mean age of 4.76 ± 0.95 year, which was further divided as >3 or ≤3 years.

**Table 3 nutrients-14-00026-t003:** Effectiveness of vitamin D supplementation on specific subscales/items of ASD Severity Measures.

First Author (Year)	ASD Severity Measure	Subscales/Items	Before-after Change (Mean Trend)	Between Groups Comparison (*p*-Value)
Feng (2017) [60]	ABC *	Sensory	Decrease	N/A
		Social skills	Decrease †	N/A
		Body and object use	Decrease †	N/A
		Language	Decrease †	N/A
		Social or self-help	Decrease †	N/A
Javadfar (2020) [58]	ABC-C subscales	Irritability	Decrease †	NS
		Lethargy/social withdrawal	Decrease	NS
		Hyperactivity	Decrease †	NS
		Inappropriate speech	Decrease	NS
		Stereotypic behavior	Decrease	NS
Jia (2019) [65]	ABC *	Sensory	Mostly Decrease *	N/A
		Social skills	Mostly Decrease *	N/A
		Body and object use	Mostly Decrease *	N/A
		Language	Always Decrease *	N/A
		Social or self-help	Mostly Decrease *	N/A
Kerley (2017) [63]	ABC subscales	Irritability	Decrease	NS
		Lethargy/social withdrawal	Decrease	NS
		Hyperactivity	Decrease	NS
		Inappropriate speech	Decrease	NS
		Stereotypic behavior	Decrease	NS
	DD-CGAS	Self-care	Increase	Increase †
		Communication	Increase	NS
		Social behaviour	Increase	NS
		School/academic	Increase	NS
Mazahery (2019) [69]	ABC domains	Irritability	Decrease	Decrease †
		Lethargy/social withdrawal	Decrease	NS
		Hyperactivity	Decrease	Decrease †
		Inappropriate speech	Decrease	NS
		Stereotypic behavior	Decrease	NS
Mazahery (2019) [68]	SRS	Social-communicative functioning	Decrease	NS
		Social awareness	Decrease	NS
		Social cognition	Decrease	NS
		Social communication	Decrease	NS
		Social motivation	Decrease	NS
		repetitive/stereotypic interests and behaviours	Decrease	NS
	SPM	Vision	Decrease	NS
		Hearing	Decrease	NS
		Touch	Decrease	NS
		Taste and smell	Decrease	NS
		Body awareness	Decrease	NS
		Balance and motion	Decrease	NS
		Social participation	Decrease	NS
Saad (2016) [55]	CARS	Relating to people	Decrease †	N/A
		Emotional response	Decrease †	N/A
		Imitation	Decrease †	N/A
		Body use	Decrease †	N/A
		Object use	Decrease †	N/A
		Adaptation to change	Decrease †	N/A
		Listening response	Decrease †	N/A
		Taste, smell, touch	Increase	N/A
		Visual response	Decrease †	N/A
		Fear	Decrease	N/A
		Verbal communication	Decrease	N/A
		Activity level	Decrease	N/A
		Non-verbal communication	Increase	N/A
		Level and consistency of intellectual response	Decrease	N/A
		General impression	Decrease †	N/A
	ABC subscales	Irritability	Decrease †	N/A
		Lethargy/social withdrawal	Decrease †	N/A
		Hyperactivity	Decrease †	N/A
		Inappropriate speech	NS	N/A
		Stereotypic behavior	Decrease †	N/A

* 3 cases, 1 month supplementation, repeated twice. † indicates significant at *p* < 0.05

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
