# Peer review of "The Role of Vitamin D Supplementation in Children with Autism Spectrum Disorder: A Narrative Review"

_nutrients, 2021, doi:10.3390/nu14010026_

Round 1

Reviewer 1 Report

Dr. Kittana et al.  evaluated the importance of Vitamin D supplementation in ASD development and the role of Vitamin D level in ASD severity. They performed a sistematic review of 12  clinical trials  studies obtained by pubmed and cochrain collection.   

They reported that improved vitamin D status significantly reduced the ASD severity, however, this effect was not consistently different between the treatment and control groups. Some limitation points were suggested possibly  explaining this results, as:  different  age of enrrolled subjects, small number of participants and a short duration of follow-ups in the selected studies.

The review is well written and addresses a very interesting point that can have a very useful implication in ASD treatment.

However, despite the authors hinting in the introduction at the importance of the genetic control of vit D receptors and Vitamin binding protein protein in the  compromised vitamin D status, they do not face this aspect in the review.

It would be of interest impoving the study by adding in the review  also studies evaluating vit D supplementation outcome in relationship with Vit D receptor end vit D binding protein genetic asset if there is any. If no valid studies on this point has been conducted,  this point may be considered a further limitation point suggesting etherogenetiy of supplementation effect in different subjects and  the authors may suggest this as furhter intersting point to be more in deeep studied.   

Minor revision: check references  at page 9 line 326

Author Response

Comment

Reply

Line #

Dr. Kittana et al.  evaluated the importance of Vitamin D supplementation in ASD development and the role of Vitamin D level in ASD severity. They performed a systematic review of 12 clinical trials studies obtained by pubmed and cochrain collection.   

The authors would like to clarify that the manuscript is a narrative review and that it did not follow all of the steps of a systematic review.

--

They reported that improved vitamin D status significantly reduced the ASD severity, however, this effect was not consistently different between the treatment and control groups. Some limitation points were suggested possibly explaining this results, as: different age of enrolled subjects, small number of participants and a short duration of follow-ups in the selected studies.

--

--

The review is well written and addresses a very interesting point that can have a very useful implication in ASD treatment.

Thank you for your positive feedback

--

However, despite the authors hinting in the introduction at the importance of the genetic control of vit D receptors and Vitamin binding protein in the compromised vitamin D status, they do not face this aspect in the review. It would be of interest improving the study by adding in the review also studies evaluating vit D supplementation outcome in relationship with Vit D receptor end vit D binding protein genetic asset if there is any. If no valid studies on this point has been conducted, this point may be considered a further limitation point suggesting heterogeneity of supplementation effect in different subjects and the authors may suggest this as further interesting point to be more in deep studied.   

Thank you for your important observation. Due to the scarcity of clinical trials on the outcome of vit D supplementation on vit D receptors and vitamin binding protein, it could not be included in the review itself. Nevertheless, such associations from findings of other types of studies were addressed in the Discussion section.

Accordingly, insufficient studies on the above aspects in context to children with ASD is highlighted as a limitation.

357-374

Minor revision: check references at page 9 line 326

Perhaps due to editing by the Editorial team, the line number has probably shifted from 326 in original version to line number 329 in the version received for revision.

This [REF] has been deleted and the actual reference was not added because these were the inferences drawn from all of the studies included in the review. 

352

Reviewer 2 Report

Introduction

Lines 56-57:

The notation suggests that 1,25(OH)D is vitamin D3 or calcitriol (?).

1,25(OH)D is not vitamin D3, but calcitriol. Vitamin D3 is cholecalciferol, formed in the skin under the influence of UVB from 7-dehydrocholesterol. It should be corrected.

Table 1. There are typos in the content of the Table (e.g. :Comaprison” or “vitmain”) - it requires corrections.

3.1 The Table in terms of presenting the results of the cited studies is illegible. Serum vitamin D levels before and after supplementation should be listed in one of the columns, respectively to the cited studies, not in the text of the chapter. The Table should not include studies in which there was no control group - they do not apply even if we only want to mark the trend resulting from the results, but instead they effectively obscure the whole. The impact on the parameters characterizing the progression of ASD should also be divided into the supplemented and non-supplemented groups. Also those in which additional supplements, apart from vitamin D, were administered, should be excluded from the presented studies.

The results of the Case study, without the control group and in the absence of data on the Treatment procedure, also do not contribute to anything.

All studies excluded from the Table can be briefly mentioned in the text. However, including such research even in outlining trends is an over-interpretation.

In the current version of the work, it is difficult to read even the authors' intentions to signal another potential role of vitamin D. As long as, in the Introduction and with the help of Figure 1, the authors quite well justify the suggestions as to a possible relationship of vitamin D with ASD, this work is in the sense of referring to published experience adds little to support or not this suggestion.

The available studies have to undergo some selection, because some of them are not well designed. You have to be aware that the level of vitamin D in the serum is influenced by many factors (obesity, metabolic disorders, diet, comorbidities) and the lack of clear information about the levels before and after supplementation means that the presented results in the current version indicate nothing at all. Although the authors discuss the weaknesses of the presented research. Such studies should possibly be accompanied by a critical commentary.

3.2. This subchapter is in contradiction with the data given in Table 1. According to the Table 1, in the report by Feng et al. (2017), the age of the respondents was not reported, and in section 3.2 the authors describe the division of the study group into age under and over 3 years.

Discussion. Lines between 268 and 315 should be moved to the Introduction as justification for the relationship between vitamin D and ASD. The Discussion chapter should discuss possible trends and differences presented in Table 1, or suggest to take into account additional factors for a more complete analysis.

Lines between 248 and 314: this fragment adds little. It follows that vitamin D has no obvious effect on the level of proinflammatory cytokines. However, the analysis of the reports is too general and it does not convince that the cited studies took into account factors influencing serum vitamin D levels and that the authors discussed the cited results exhaustively. For example, the authors write: “(…), three recent systematic reviews concluded that vitamin D does not influence IL-6 levels significantly in obese and overweight individuals, adults, and individuals with type 2 diabetes mellitus.” The fact that there is no effect on the level of cytokines in the above-mentioned obese or type 2 diabetic people does not exclude the correctness of the hypothesis about vitamin D modulating inflammatory processes. So, the description of the cited studies in this fragment should be more precise.

Conclusion. This chapter is ok. However, the evidences for safety and tolerability of the supplements across all trials were not mentioned in the text of the paper.

Summary:

Admittedly, the authors admit that the presented research cannot be the basis for drawing specific conclusions and that further research is necessary. They also point to the weaknesses of individual experiences. However, in the opinion of the reviewer, they should not be exempt from the necessity of a critical approach to the research they cite, in the sense of selecting the examples used in the work. Studies that are unacceptable already at the design stage (no control group, no information on the treatment procedure, or a one-time response measured in a single patient) should not be used as material for any discussion of the results. It is then easy to lose any real evidence supporting the adopted thesis.

Author Response

Comment

Reply

Line #

Introduction Lines 56-57: The notation suggests that 1,25(OH)D is vitamin D3 or calcitriol (?).

1,25(OH)D is not vitamin D3, but calcitriol. Vitamin D3 is cholecalciferol, formed in the skin under the influence of UVB from 7-dehydrocholesterol. It should be corrected.

Fixed- deleted vitamin D3

57

Table 1. There are typos in the content of the Table (e.g.: Comaprison” or “vitmain”) - it requires corrections.

Addressed

Tables 1+2

3.1 The Table in terms of presenting the results of the cited studies is illegible. Serum vitamin D levels before and after supplementation should be listed in one of the columns, respectively to the cited studies, not in the text of the chapter. The Table should not include studies in which there was no control group - they do not apply even if we only want to mark the trend resulting from the results, but instead they effectively obscure the whole. The impact on the parameters characterizing the progression of ASD should also be divided into the supplemented and non-supplemented groups.

1.     Serum vitamin D results have been added.

2.     We felt it was of significance to show the results of all studies in an organized manner for easier comparison, despite their methodological quality. However, we separated the Tables based on whether the study had a control group or not. These are now presented in Tables 1 and 2 respectively.

3.     To differentiate between the impact, we have indicated whether it was significant within the same group, and between groups. Whether the results was significant or not would show whether the impact in the supplemented group matched that of supplemented or not. We have also created a new table (Table 3), which highlights the result of each subscale, to summarize the effect of vitamin D on different behaviors. 

Tables 1+2+3

Also those in which additional supplements, apart from vitamin D, were administered, should be excluded from the presented studies.

Any study without a vitamin D only supplementation group was not included in this review. The presented data in the tables would refer to the vitamin D only group if the study had groups of vitamin D + additional intervention. Between group comparisons is only done for the placebo group or no supplement group. This is clarified below the table, and the placebo group is now highlighted.

Tables 1+2

The results of the Case study, without the control group and in the absence of data on the Treatment procedure, also do not contribute to anything. All studies excluded from the Table can be briefly mentioned in the text. However, including such research even in outlining trends is an over-interpretation.

1.     Instead to total exclusion, these were presented in a different table to highlight their results. Despite their methodological quality, these results would still indicate a potential effect of vitamin D supplementation, which would warrant further investigation.

Tables 1+2

2.     Statements following “a trend’ were added to indicate careful interpretation of the results.

198-202

In the current version of the work, it is difficult to read even the authors' intentions to signal another potential role of vitamin D. As long as, in the Introduction and with the help of Figure 1, the authors quite well justify the suggestions as to a possible relationship of vitamin D with ASD, this work is in the sense of referring to published experience adds little to support or not this suggestion.

Based on the following comments, some sections were moved from the discussion to the introduction to provide an overall view of the possible relationship of vitamin D.

62-63, 65, 69-75, 83-97, 104-112

Figure 1 remained in the discussion to connect these concepts from the introduction with the outcomes presented in the results section.

The published literature is limited in regards to supporting an exact pathway linking vitamin D with ASD; therefore these were postulated and suggested for future research in Figure 1.

Figure 1

The available studies have to undergo some selection, because some of them are not well designed. You have to be aware that the level of vitamin D in the serum is influenced by many factors (obesity, metabolic disorders, diet, comorbidities) and the lack of clear information about the levels before and after supplementation means that the presented results in the current version indicate nothing at all. Although the authors discuss the weaknesses of the presented research. Such studies should possibly be accompanied by a critical commentary.

1.     Levels before and after supplementation were added to aid in the interpretation of the results

Tables 1+2.

2.     Some factors that could influence vitamin D levels were clarified in the limitation (no change), variations in ASD assessment tools and genetic polymorphisms were added as a possible contributor to the vitamin D status.

352-374

3.2. This subchapter is in contradiction with the data given in Table 1. According to the Table 1, in the report by Feng et al. (2017), the age of the respondents was not reported, and in section 3.2 the authors describe the division of the study group into age under and over 3 years.

Feng et al. initially recruited a larger number of children with ASD, and only reported their mean age (no age range as part of the inclusion criteria was described). The authors later chose a smaller group for the supplementation trial, and no further info on the demographics was provided, only that they were divided into two groups (<=3 or >3). A brief description is added in the footnote for clarification.

Table 2

Discussion. Lines between 268 and 315 should be moved to the Introduction as justification for the relationship between vitamin D and ASD.

Some sections were moved from the discussion to the introduction.

The discussion section was rearranged.

The figure remained in the discussion to link VD’s association with ASD presented in the introduction with the results presented in section 3. The dosages clarified in the figure are now restricted to the doses used in studies with control groups.

62-63, 65, 69-75, 83-97, 104-112, Figure 1

The Discussion chapter should discuss possible trends and differences presented in Table 1,or suggest to take into account additional factors for a more complete analysis.

The discussion section was rearranged by merging the main results followed by the discussion immediately under the same subheading. Some sections were re-phrased and more discussion was added to support these sections.  The sections highlighted in green were moved from section 4 and merged with section 3

198-211, 255-257, 287-289, 299-313, 319-325.

Lines between 248 and 314: this fragment adds little. It follows that vitamin D has no obvious effect on the level of proinflammatory cytokines. However, the analysis of the reports is too general and it does not convince that the cited studies took into account factors influencing serum vitamin D levels and that the authors discussed the cited results exhaustively. For example, the authors write: “(…), three recent systematic reviews concluded that vitamin D does not influence IL-6 levels significantly in obese and overweight individuals, adults, and individuals with type 2 diabetes mellitus.” The fact that there is no effect on the level of cytokines in the above-mentioned obese or type 2 diabetic people does not exclude the correctness of the hypothesis about vitamin D modulating inflammatory processes. So, the description of the cited studies in this fragment should be more precise.

As discussed, part of this section was moved to the introduction, and other parts were merged with section 3.

62-63, 65, 69-75, 83-97, 104-112, 255-257, 287-289, 299-302,  319-323.

The discussion on inflammatory markers was modified. Other factors may influence this association, especially the high heterogeneity of studies included in the systematic reviews. The aim of this was to present an inconsistency in the literature, and propose future studies to focus on this association in ASD as a mediating factor between VD and severity of symptoms.

299-313

A statement was added after the serotonin discussion to highlight that no conclusions can be made regarding ASD.

323-325

Conclusion. This chapter is ok. However, the evidences for safety and tolerability of the supplements across all trials were not mentioned in the text of the paper.

A section on safety and tolerability was added

326-339

Admittedly, the authors admit that the presented research cannot be the basis for drawing specific conclusions and that further research is necessary. They also point to the weaknesses of individual experiences. However, in the opinion of the reviewer, they should not be exempt from the necessity of a critical approach to the research they cite, in the sense of selecting the examples used in the work. Studies that are unacceptable already at the design stage (no control group, no information on the treatment procedure, or a one-time response measured in a single patient) should not be used as material for any discussion of the results. It is then easy to lose any real evidence supporting the adopted thesis.

Thank you very much for the feedback on this article.

It is true that the inclusion criteria was not strict regarding the methodological quality, however this was mainly due to the small number of trials published on this topic. Given the important physiological role VD may play in ASD, the authors wished to provide an overall view of the studies that are available in the literature, however with careful interpretation.

To avoid comparison of results between studies with and without control groups, these were separated into two tables, as previously elaborated.

--

Round 2

Reviewer 1 Report

The manuscript has been largely improved. It is very interesting  and opens new perspectives for future studies 

Reviewer 2 Report

You should appreciate the work put into proofreading the work. It was mostly prepared in accordance with the reviewer's recommendations.

However, the authors should still make some corrections, according to the following points:

  1. Table 1:

 - The column titles should be standardized on pages 6 and 7/27, i.e.: “25(OH)D change in controls (ng/ml)” or “25(OH)D levels (ng/ml)/Controls”. Similarly in the columns dedicated to the cases;

 - Pos. 6 - how to explain the note as follows: 6.48 +/- 24.36 (p <0.05)?

 - Pos. 7- please, present in more clear way the note as follows: “25(OH)D <15: 5000 IU/day”; “25(OH) D 15-20: 400 IU/day”; “25(OH)D is 15-20: continue”.

  1. In the reviewer's opinion, Table 3 is not necessary - the results described in are repeated in Table 1 and 2. These Tables (1 and 2) are sufficient to draw the same conclusions as described.
  2. The title of section 3.1 and Table 1: “Vitamin D status and associations with ASD severity” should be revised - it suggests that the section 3.1 and Table 1 concern a correlation between baseline calcidiol levels and ASD advancement, while in fact they illustrate the impact of vitamin D supplementation on the picture of this disease. Therefore, the reviewer suggests that the title of subsection 3.2 should become the title for 3.1 and that both subsections be combined into one. Then, the numbering of the subsequent ones will change as follows: 3.3 into 3.2 3.4 into 3.3 and 3.5 into 3.4.